# **Retrieval of effective aerosol diameter from satellite observations**

Humaid Al Badi<sup>1,3</sup>, John Boland<sup>1</sup>, David Bruce<sup>2</sup>

<sup>1</sup>Centre for Industrial and Applied Mathematics and <sup>2</sup>Natural and Built Environments Research Centre, University of South Australia, Mawson Lakes, 5095, Australia.

<sup>3</sup> Directorate General of Meteorology, Public Authority for Civil Aviation, Oman.

Correspondence to: Humaid Al Badi (h.albadi@gmail.com)

Abstract. Dust aerosol particle size plays a crucial role in determining dust cycle in the atmosphere and the extent of its impact on the other atmospheric parameters. The in-situ measurements of dust particle size are very costly, spatially sparse
 and time-consuming. This paper presents an algorithm to retrieve effective dust diameter using infrared band brightness temperature from SEVIRI (the Spinning Enhanced Visible and InfaRed Imager) on the Meteosat satellite. An empirical model was constructed that directly relates differences in brightness temperatures of 8.7, 10.8 and 12.0 µm bands to dust effective diameter using the Mie extinction efficiency factor. Three case studies are used to test the model. The results showed consistency between the model and in-situ aircraft measurements. A severe dust storm over the Middle-East is

15 presented to demonstrate the use of the model. This algorithm is expected to contribute to filling the gap created by the discrepancies between the current size distributions retrieval techniques and aircraft measurements. Potential applications include enhancing the accuracy of atmospheric modelling and forecasting horizontal visibility and solar energy system performance over regions affected by dust storms.

#### 20 **1. Introduction**

Aerosols including dust have a significant impact on the climate through a range of complex mechanisms. In the short term, the effects of aerosol variability generate perturbations in atmospheric turbidity. Aerosols alter atmospheric turbidity by modifying the flux of solar short-wave and terrestrial long wave radiation through scattering and absorption (Goudie & Middleton 2001). The amount of absorption and forward and backward scattering depends on the concentration, size

- distribution and chemical composition of aerosol particles. Despite continued research, the multiple aerosol effects are still poorly represented in climate models which lead to substantial uncertainty (Ben-Ami et al. 2010; Boucher et al. 2013). One of the reasons for such uncertainty might be the scarcity of adequate and routine measurements of aerosol properties. Most of the particle size in-situ measurements in operation take place on the ground by sampling the precipitated aerosols (Afeti & Resch 2000; Sunnu, Afeti & Resch 2008). Ground measurements are not sufficient because aerosols have different and dynamic vertical distribution. Limited aircraft campaigns have been conducted to sample the aerosols particle size in the
- atmosphere ( e.g. Tanré et al. 2003; Müller et al. 2012; Ryder et al. 2013b; Ryder et al. 2013a).

The sparsity of in-situ dust sampling opened the door for remote sensing techniques to fill the gap. Nakajima et al. (1996) and Dubovik & King (2000) developed inversion algorithms for retrieval of the aerosol volume distribution  $\left(\frac{dV}{d\ln r}\right)$  from Sun and sky radiance ground measurements, such as AErosol RObotic NETwork (AERONET). From space, Atmospheric Infrared Sounder (AIRS) observations made at 9.4 µm are used to retrieve dust effective radius (Pierangelo et al. 2005). Klüser et al. (2011) also used a Singular Vector Decomposition method on observed spectra from the Infrared Atmospheric Sounding Interferometer (IASI) to extract dust effective radius. However, the current techniques do not satisfy the need for accurate dust particle size data. Discrepancies have been observed in retrieving the size distributions between the AERONET algorithm (Dubovik & King 2000), the Sky Radiation

comparison between in-situ sampling during Saharan Mineral Dust Experiment (SAMUM) 2006 and AERONET effective radii retrieval showed that the Sun photometer observations are smaller by a factor of two compared to the in-situ observations (Müller et al. 2012). The underestimation of the dust size distribution also appears to be a common pattern among the current satellite algorithms (Klüser et al., 2011; Pierangelo et al. 2005). The abundance of large particles over desert surfaces might have a role in reducing the accuracy of the current sunphotometer and satellite techniques (Ryder et al.

(SKYRAD) algorithm (Nakajima et al. 1996) and aircraft measurements (Estellés et al. 2012; Ryder et al. 2015). The

15 2013b). This paper presents a new approach in retrieving effective dust particle size to reduce the gap between the observed dust particle size and the current retrieval methods.

The algorithm uses infrared band brightness temperature from SEVIRI (the Spinning Enhanced Visible and InfaRed Imager) on Meteosat satellite. Infrared window bands 12.0, 10.8 and 8.7 µm are widely used to track dust plumes in operational meteorological applications. In 2003 the first METEOSAT Second Generation Satellite was launched carrying SEVIRI. It has an unprecedented temporal resolution of 15-minutes over the Sahara Desert and the West Asia regions where the primary dust sources in the world are located. The Dust Red, Green and Blue (Dust RGB) image composite corresponding to infrared window band combination of 12.0-10.8, 10.8-8.7 and 10.8 µm respectively, is one of the most used combinations to track dust clouds (EUMETSAT 2016). The Dust RGB composite uses the fact that the change of the bands brightness

- Temperature (T) is strongly correlated to the change in the scattering and absorption caused by dust variability in the atmosphere. That is, the dust aerosol variability alters the radiation flux falling on the satellite radiometer from which T is calculated. The correlation between T and dust aerosols is rather complex and linked to many parameters. It is mainly caused by Aerosols Optical Depth(AOD), dust particle size and shape and the emissivity which in turn linked to dust chemical composition (e.g. Brindley et al. 2012; Klüser et al. 2011). The method presented here shows it is possible to reduce this
- 30 correlation complexity when the brightness temperature difference is used to retrieve effective dust size. An analytical approach of the Mie extinction efficiency factor is used to build an empirical model to link effective dust diameter with brightness temperature difference of 8.7 and 12.0 μm.

#### 2. Calculating the Mie scattering efficiency factor for dust

To assess how much a spherical dust particle scatters light, the extinction efficiency factor  $Q_{ext}$  needs to be introduced. Any particle of diameter *d* which blocks the radiation path will remove power from the incident radiation with intensity  $I_0$  by an amount  $P_{removed}$ 

5

$$P_{removed} = C_{ext} I_0$$

where  $C_{ext}$  is the extinction cross section (Hahn 2009).

Mie's solution for Maxwell's electromagnetism equations results in:

$$C_{ext} = \frac{\lambda^2}{2\pi} \sum_{n=1}^{\infty} (2n+1) Re\{a_n(x,m,\Psi_n,\xi_n) + b_n(x,m,\Psi_n,\xi_n)\}$$

where  $x = \frac{\pi d}{\lambda}$  and it is called the size parameter, *m* is the refractive index,  $a_n(x, m, \Psi_n, \xi_n)$  and  $b_n(x, m, \Psi_n, \xi_n)$  are Mie 10 scattering coefficients derived from solving Maxwell's equations, and  $\Psi$  and  $\xi$  are the Ricatti-Bessel functions (Hahn 2009). The extinction efficiency factor  $Q_{ext}$  is commonly used to relate the extinction cross section  $C_{ext}$  to the geometrical cross section  $C_{geo}$ :

$$Q_{ext} = \frac{C_{ext}}{C_{geo}}$$

For a sphere  $C_{geo} = \frac{\pi d^2}{4}$ , Thus:

15

$$Q_{ext} = \frac{2}{x^2} \sum_{n=1}^{\infty} (2n+1) Re\{a_n(x,m,\Psi_n,\xi_n) + b_n(x,m,\Psi_n,\xi_n)\}$$

This equation is solved numerically for any given x and m (Hahn 2009).

20

The refractive index is wavelength and chemical composition dependent. In this paper, the Di Biagio et al. (2014) estimation of the refractive index *m* has been used where,  $m_{8.7} = 1.10 + 0.20i$ ,  $m_{10.8} = 1.9 + 0.25i$ ,  $m_{12.0} = 1.75 + 0.40i$  are the values given for the 8.7, 10.8 and 12.0 µm refractive index respectively. These refractive index estimations were made in the laboratory for five dust samples collected during dust events originated from different Western Saharan and Sahelian areas (Di Biagio et al. 2014). Figure 1 shows the calculated Mie extinction efficiency factor  $Q_{ext}$  for particle diameter from 1 to 50 µm using MiePlot software (Laven 2016). Berg et al. (2011) provide an explanation why  $C_{ext} \rightarrow 2C_{geo}$  and  $Q_{ext} \rightarrow 2$  when the particle diameter becomes very large.

#### 3. Deriving the model

As Figure 1 shows, Mie theory predicts a significant change in the extinction efficiency factor of the thermal infrared 12.0, 10.8 and 8.7  $\mu$ m when the particle diameter lies between 1 and 20  $\mu$ m. This dust range covers the reported effective dust particle size range during the Fennec 2011 aircraft dust sampling campaign over West Africa which was between 2.3 to 19.4  $\mu$ m for some dust events (Ryder et al. 2013b).

Since  $C_{ext} = \frac{P_{removed}}{I_0}$  then  $Q_{ext} \alpha \frac{1}{I_0}$ . But  $T \alpha I_0$ ; thus  $Q_{ext} \alpha \frac{1}{T}$ . This implies that the peaks of  $Q_{ext}$  correspond to troughs in brightness Temperature (T). Thus, a small change in effective dimeter d in the range 2.3 to 19.4 µm will result in a significant change in brightness temperature. The pattern of 8.7 and 12.0 µm curves in Figure 1 also suggests that the difference  $T_{8.7-12.0}$  versus effective diameter d will have the same curve shape as properly scaled Ryleigh distribution function. The Ryleigh distribution has the generalised formula:

10

30

5

$$f(x) = \frac{x}{\alpha^2} e^{-\frac{x^2}{\alpha^2}} \quad (Walck \ 2007)$$

Thus

 $T_{8.7-12.0} \ \alpha \ a' \frac{d}{b'^2} \ e^{-\frac{d^2}{b'^2}}$  where a', b', c' are scaling coefficients

The theoretical curves of extinction coefficient in Figure 1 assumes the same amount of energy emitted from the ground
towards space. In reality, there is a significant difference in the ground emissivity between the three bands; 12.0, 10.8 and
8.7 μm. In this paper, the Global Infrared Land Surface Emissivity Database has been used (Seemann et al. 2008). As
Figures 2 & 3 show, the band 8.7 μm has the strongest variation in emissivity while 12.0 μm is homogenous around a
relatively high value of 0.93 over desert surfaces. Assuming 12.0 μm emissivity is constant, as 8.7 μm emissivity (€<sub>8.7</sub>)
increases, the difference T<sub>8.7-12.0</sub> also increases. Additional impact of emissivity difference originates from the dust layer.
Furthermore, as the dust diameter increases the contribution of the dust layer emissivity increases (Takashima & Masuda 1987). In summary, it is safe to say;

$$T_{8.7-12.0} \ \alpha \ \in_{8.7}^2 d$$

which leads to:

$$\frac{T_{8.7-12.0}}{\epsilon_{8.7}^2} = a \frac{d^2}{b^2} e^{-\frac{d^2}{b^2}} + c d + f$$

25 where  $\in_{8.7}$  is the ground emissivity at 8.7 µm; *a*, *b*, *c* and *f* are constants.

Actual values of  $\left(\frac{T_{8,7-12.0}}{\epsilon_{8,7}^2}\right)$ , *d*) from two dust events were used to calculate the coefficients *a*, *b*, *c* and *f* numerically. The first one is a dust case sampled by Fennec b604, 20 June 2011 (Ryder et al. 2013b). The reported mean *d* was around 6 µm and the emissivity  $\epsilon_{8,7}$  at the location (Figure 4) is 0.72. Figure 5 shows the brightness temperature change of 8.7, 10.8 and 12.0 µm bands versus time of the 20<sup>th</sup> of June 2011 at the experiment location (24.0N, 10.0W). The second dust event was a

Atmospheric §

severe dust storm which is utilized to guide the model at high *d* values. Figure 6 shows the brightness temperature of a severe dust case over West Asia at location 27.0N, 47.8E (detailed description is in Section 5). In this case, few points of  $\left(\frac{T_{8.7-12.0}}{\epsilon_{8.7}^2}\right)$ , *d* could be estimated from the small pocket of  $T_{12.0-10.8}$  negative values formed around 15 µm diameter when  $T_{8.7} - T_{12.0} > 0$ . That occurs when 10.8 µm band extinction factor is less than 12.0 µm band extinction factor which is around 15 µm in Figure 1.

(Figures 4, 5 & 6 here)

The numerical solution results in coefficients: a = 29, b = 12.5, c = 1 and f = -29.2. Figure 7 shows a plot of the model  $\frac{T_{8.7-12.0}}{\epsilon_{8.7}^2}$  versus the effective diameter *d* in the range [1, 25] µm with the actual points used to calculate the coefficients *a*, *b*, *c* and *f*. For convince, a Look Up Table (LUT) was created for given *d* and the corresponding  $\frac{T_{8.7-12.0}}{\epsilon_{8.7}^2}$  values.

10

15

5

### (Figure 7 here)

Although the extinction efficiency in Figure 1 implies that  $T_{12.0}$  should be greater than  $T_{10.8}$ ; in a low dust concentration atmosphere  $T_{12.0} 

### $\odot$

#### Testing the algorithm 4.

Three cases over West Africa (Figure 10) are presented here to test the algorithm. The sampling in the three cases was carried out by Fennec aircraft campaign during June 2011 over West Africa (Ryder et al. 2013b). In all cases, the location of brightness temperature curves was chosen to be as close as possible to the middle of the sampling area and where there was minimum cloud presence at the time of sampling. The time slots with cloud contamination have been removed.

(Figure 10 here)

#### 4.1. Case 1: Mali; 17-18 June 2011:

The dust was sampled by Fennec flight number b600, 17 June 2011 10:00 to 11:15 UTC during the emission phase of the 10 dust event (Ryder et al. 2013b). There was another sampling mission (Fennec 601) on the same day between 17:15 to 18:15 UTC. The reported mean d from Fennec sampling was around 12.3 µm. Figure 11 shows the calculated d using the model and the BT of 8.7, 10.8 and 12.0 µm bands versus time at 21.2N, 5.6W. The 8.7 µm ground emissivity in the location is 0.712 using the Global Infrared Land Surface Emissivity Database (Seemann et al. 2008). There was enough dust concentration for this method to be used starting from 12:00 UTC. The average d from 12:00 to 18:00 UTC was found to be 9.6 µm. This value is expected to be less than the one reported by Fennec sampling for a "recent uplift" dust event. In the 15

- recent uplift stage, incoherent structure of the dust cloud is at maximum, where large particles of dust are present in lower levels and fine dust in the higher level. Another good reason that might contribute to the underestimation is the Fennec aircraft sampling method. The sampling was limited to altitudes beneath 2400m above the ground level while SEVIRI measures the radiation coming from the upper part of the dust cloud which might have smaller dust size at higher altitudes. (Figure 11 here)
- 20

25

5

### 4.2. Case 2: Mauritania; 25 June 2011:

The second case is another "recent uplift" dust emission. The case was sampled by Fennec b610, 25 June 2011 09:15 to 10:45 UTC (Ryder et al. 2013b). The sampled mean d was around 8.6  $\mu$ m. Figure 12 shows the calculated d and the BT of 8.7,10.8, 12.0 µm bands versus time of the 25<sup>th</sup> of June 2011 at the location (25.8N, 7.4W). Emissivity of 8.7 µm band at the location is 0.712. The average d between 0800 to 1130 UTC is calculated to be 6.2  $\mu$ m which is again expected given the low level sampling which probably selected larger particles due to inhomogeneous fresh dust cloud as in Case 1.

(Figure 12 here)

#### 4.3. Case 3: Mauritania; 24-26 June 2011:

30

This case is a case of long transported dust and covers a relatively large area which was sampled by four Fennec flights missions over three days (Ryder et al. 2013b). The emissivity of 8.7 µm band at the location is 0.732. Figure 13 shows the calculated d and the brightness temperature of 8.7,10.8, 12.0  $\mu$ m bands versus time of the 25<sup>th</sup> to 26<sup>th</sup> of June 2011 roughly

in the centre of the sampling area (23.7N, 10.3W). In this case the sampled and retrieved effective diameter d showed very good agreement. The average sampled d for the three days around the flight hours was 5.9 µm while the model retrieval shows d of 6.0 µm.

#### (Figure 13 here)

5 Table 1 provides a summary of the testing results. Despite underestimation in recent uplift cases, overall, the model gave promising results. In cases 2 and 3 the sampled value was within 95% confidence interval for a single value and in case 1 it was just outside this interval.

(Table 1 is here)

#### 5. Use of the algorithm

- 10 Potential applications for the model include:
  - a. Verification of atmospheric aerosols models. This application is crucial because of the scarcity of airborne aerosol in-situ measurements.
  - b. Horizontal visibility forecasting. A sudden drop in horizontal visibility during dust storms is known to be the most direct and hazardous effect of dust storms. Since horizontal visibility is particle diameter dependent, combining particle diameter data from this model with the carrying air mass trajectory forecast from atmospheric models can give an indication of the horizontal visibility from a few hours to a couple of days depending on the location of the emission source.
  - c. Solar energy system performance forecasting. The performance of the solar power systems depends on the turbidity of the atmosphere which has a correlation with the effective particle diameter. The technique can give an indication of the amount of dust that will precipitate on solar energy systems from an upcoming dust event.
  - d. Assist in studying the transport behaviours of dust and volcanic ash in the atmosphere.

A severe dust storm is presented here as an example of the model use. The aim is to check model behaviour in severe cases and how dust particle size will change over an extended period. The dust storm originated on  $1^{st}$  of April 2015 over the Arabian Peninsula and affected a large area of western Asia. The brightness Temperature *T* of the three SEVIRI bands and

25 effective diameter retrieval *d* was plotted against time around the dust cloud passage for three locations along the track of the dust cloud movement.

Location # 1 was chosen to be close to the emission source and downstream of wind flow to pick the maximum concentration of emitted dust. The location is at around 300 km southeast the centre of the emission source (Figure 14).

#### (Figure 14 here)

#### (Figure 15 here)

The average background aerosol effective diameter in the early hours of  $3^{rd}$  of April is calculated to be 6.7 µm which is not far from the reported background dust of 7.2 by Fennec aircraft campaign during June 2011 (Ryder et al. 2013a). The slight difference can be explained by the heat low pressure, that develops during summer over the desert and helps to keep larger

Atmospheric Measurement Techniques

dust particles longer in the air through dry convection. The maximum d in this case was around 14.0  $\mu$ m (Figure 15) which is within the range of the reported d by Fennec aircraft campaign (2.3 -19.4  $\mu$ m).

The effective diameter d can be represented in a 2D map. Figure 16 shows an example for 1st April 2015 18:15 UTC. Most of the clouds were screened out; however, a few water clouds still manifest themselves in this product (e.g. Southeast coast of Yemen). The use of a sophisticated cloud screening algorithm could improve this aspect of the results.

#### (Figure 16 here)

Location #2 is the city of Abu Dhabi (Figure 17) and correspond to 24 hours later than Figure 14. Figure 18 shows that the maximum calculated d has dropped to 12.3 µm from 14.0 µm at Location#1 24-hour prior. This drop coincides with the fact that d in a dust cloud is inversely proportional with time, because as time progresses large dust particles are precipitated

leaving smaller particles in suspension.

#### (Figure 17 here)

#### (Figure 18 here)

Location #3 is chosen to envisage the evolution of the effective diameter of long transported dust after three days from emission (Figure 19 and Figure 20). Xu et al. (2010) found that the volume average diameter of dust particles coming from

15 the sources in western Asia ranged between 3.2 to 4.2  $\mu$ m over the central Himalaya. For Location#3 in the research reported here and following several cloud animations, on average, the air mass carrying the dust needs around 5.5 days to move from a source over the centre of the Arabian Peninsula and to the central Himalaya. The calculated three-hour average of *d* after three days was 10.1  $\mu$ m between 06 to 08 UTC on the 4<sup>th</sup> April. Although central Himalaya is outside SEVIRI coverage , there are still 2.5 days to *d* to reduce to the average diameter presented by Xu et al. (2010).

> (Figure 19 here) (Figure 20 here)

5

# 6. Potential future improvement in the model

25

The accuracy of the numerical solution for the coefficients a, b, c and f in the model can be improved if more in-situ data were used in the calculation. This is important to dilute the bias made by individual in-situ measurements. An example of the bias that could be avoided is the one resulting from constraining the sampling to altitudes lower than 2400 meters, with most samples acquired in a lower part of this air mass (Ryder et al. 2013b). The sampling will be more representative of the column average - from the satellite perspective- if it is extended to higher levels and if sampling time were more evenly distributed vertically. There is also lack of intense dust storms in the published sampled data, with most aircraft sampling

30 being undertaken during relatively low to moderate dust emission events. This is probably for safety reasons, but it does limit the validation of the method for major dust events with larger particle sizes. It will be interesting to observe the use of emerging drone technology to sample dust in intense to severe dust storms. Such data should help to clarify many aspects of dust storms dynamics in general and, fine-tune this model in particular.

Future work will include testing the model with another satellite radiometer outside SEVIRI coverage area. One candidate is the new Advanced Himawari Imager (AHI) on board the Himawari-8 satellite. This instrument provides data that potentially can be exploited to retrieve effective diameter for dust clouds over Australia and central/east Asia. Another interesting feature in Himawari-8 AHI is its extra spectral band in the thermal infrared range. In principle, with more spectral bands, the accuracy of retrieval should increase especially in respect to the larger dust size.

10

#### 7. Conclusions

storm prevalence.

Dust cycle is an important part of the earth system. The current in-situ sampling data of dust particle size are sparse and expensive. Thus, remote sensing retrieval methods have an important role in covering the gap. In this paper, an empirical algorithm has been presented to estimate effective aerosol diameter d using satellite-based observations. The infrared brightness temperature of SEVIRI bands 8.7, 10.8 and 12.0  $\mu$ m were used. The algorithm showed promising consistency with the other means of estimating d in the literature (Table 1). The accuracy of estimating the coefficients in the empirical

model is expected to improve if more in-situ d measurements are used in the numerical solution. The foreseen applications include verification of atmospheric aerosols models. Furthermore, the model can assist in predicting atmospheric turbidity

when used with air-mass trajectory forecasting and hence predicting of solar energy performance in regions with high dust

9

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

Takashima, T & Masud