# Peer review of "Retrieval of effective aerosol diameter from satellite observations"

_Atmospheric Measurement Techniques, 2016_

## Referee Comment (RC1) · Anonymous Referee #3 · 15 Sep 2016

This paper uses Mie theory in an attempt to relate aerosol (specifically mineral dust aerosol) size to brightness temperature differences (BTDs) in the thermal infrared measured by the Spinning Enhanced Visible and InfraRed Imager (SEVIRI). The authors use aircraft observations from the Fennec campaign to create an empirical relationship between one of the BTDs and effective dust particle size before comparing other cases from Fennec with values retrieved using their empirical model.

Overall, while I think the aim is laudable I find the approach wholly unconvincing, supported as it is by incomplete physics.

In my opinion there are a number of serious weaknesses described below:

(i) The radiative transfer theory used here is far too simplistic. In reality, in the thermal infrared, one needs to consider both extinction (scattering and absorption) by, and

emission from, the dust layer. Moreover, one has to consider emission from the surface and the underlying (and possibly overlying) atmosphere dependent on the dust height (atmospheric temperature structure) and the wavelength considered if one is to correctly interpret the satellite signal.

(ii) There seems to be no appreciation of the Beer-Lambert law, or alternatively the differential nature of extinction, as radiation passes through a medium. Just considering absorption and removal by scattering by the dust layer alone would lead to an exponential dependence of the final 'intensity' on optical depth, which is itself a function of the extinction cross section. Add in emission, plus scattering into the upward direction and you will obtain the full radiative transfer equation (usually expressed in radiance although conversion to irradiance is possible if done properly).

(iii) The various relationships given near the start of section 3 are hence not correct. In fact, even if the earlier assumptions were ok I can't see how they would logically follow. Why should the extinction efficiency be inversely proportional to the radiation incident on the dust layer? The former is an intrinsic property of the dust and is only dependent on the size distribution, shape of particles and composition. Similarly, brightness temperature is not directly proportional to the radiation incident on the dust. It is not even directly proportional to the intensity (as defined here) on the satellite radiometer but rather results from a non-linear conversion of the incident radiance using the Planck function.

(iv) As noted above, aerosol optical properties are related to composition, shape and size distribution. The use of Mie theory as given implicitly assumes that the particles are spherical which is rather unlikely for dust. Moreover, the authors simply use one set of refractive indices yet compare a number of different cases, including African and Arabian dust events. One might anticipate significantly different compositions dependent on source. While the assumption of sphericity is likely to be less severe in the IR than the visible, at the very least some sort of sensitivity analysis should be performed to assess the impact of uncertainty in the dust composition on the resulting BTDs.

(v) As written, it is difficult to see whether the authors have any concept of the effect of a size distribution. Their Mie calculations appear to have been carried out for single particles (although I am not sure of this as the 'ringing' that one might expect to see in this case is absent). In reality, these responses will be weighted by the fraction of particles within each size bin, which will vary from dust event to dust event (and even during an individual dust event). Hence, when looking at real signals, the shapes in figure 1 will effectively be distorted differently for different distributions of particles such that fitting one empirical model is unlikely to be representative.

(vi) Similarly, have the authors taken the spectral width of the SEVIRI channels into account? It is not clear from what has been written. Since the filters are quite wide they will also affect the size of the signal seen and its variability. The viewing angle of the satellite will also affect the signals seen due to differential absorption through the atmosphere.

(vii) In the derivation of their model the authors appear to make the assumption that the dust plume emissivity is the same as the surface emissivity (at least this is how it reads to this reviewer). This is not valid as, even if the composition is the same, the lofted particles are likely to be smaller and less densely packed than those at the surface.

(viii) It is totally unclear where the 'measurements' at 15 micron used to fit the model have come from. In any case, using two clustered points to perform a curve fit such as that shown in figure 7 is, in my opinion, very bad science. I could fit any line I wanted through those points.

(ix) Although uncertainties are given in table 1 it is unclear exactly how these have been derived. Are these propagated correctly through the model (e.g. do the authors consider the effects of uncertainties in their fit, let alone those in surface emissivity, composition etc.)?

In summary it should be noted that this reviewer is unconvinced that, even using the correct radiative transfer theory, and employing the simplifying assumptions that dust

particles are spheres and have the same composition everywhere, there is enough independent information in a single BTD to extract size information. To do this I would suggest that at the very least, dust optical depth and height need to be known, and even then one would still have to account for the impact of confounding influences such as variable surface temperature, surface emissivity and water vapour content. It may be that there are 'regimes' of behaviour (e.g. dust plumes above a certain optical thickness) where size information can be extracted but I suggest the authors perform a much more comprehensive suite of (correct) radiative transfer calculations (explicitly simulating SEVIRI BTDs, including the relevant instrument characteristics) to look at whether what they are attempting to do is actually feasible. If they believe it is then they also need to come up with a much more convincing strategy for validating their results, including a traceable uncertainty analysis.

---

## Referee Comment (RC2) · L. Klüser (Referee) · 19 Sep 2016

I cannot recommend publication of the manuscript in the current form. I have the impression that the manuscript was written quite hastily, but also that several oversimplifications have been made without appropriate justification. I absolutely acknowledge the validity of simplifications and statistical representations in the treatment of radiative transfer, especially in the context of standalone satellite retrieval methods. Nevertheless the claims presented in section 3 seem to be too drastic. At several occasions the authors claim proportionality of variables, which is not justifiable in the presented form. For example the surface emissivity is by no means proportional to the brightness temperature differences. Of course the BTD is a function also of the spectral surface emissivity, but in a highly non-linear way, even in the simplemost representation by Planck emission and radiative transfer described with the Schwarzschild equation.

Consequently the claim that the ratio of BTD and surface emissivity at $8.7\mu$m can be represented by an arbitrary distribution function (which has been introduced without any motivation of its use) cannot be physically sound. Still there might be good reasons to follow that approach, but then I would expect the authors to concisely describe the simplifications made and why they nevertheless use this statistical method. There are good examples of such approaches being successful, but the manuscript in the present form could easily yield the impression that the authors lack understanding of radiative transfer (which I do not believe). The coefficients for the solution of the statistical approach seem to fall out of the sky, the authors present no description of how they have obtained them. By solving a regression scheme? By a least-squares fit? By correlation methods? Several other issues (see specific comments below) support the impression that this study lacks the quality to be published in AMT. I fully acknoledge that the results presented by the authors look very promising and that retrieving dust particle size from space is of high importance. So I would encourage the authors to carefully read the comments and to provide a revised manuscript in response. This then has to undergo review once again. I see the potential in the authors work, that is why I am not suggesting rejection. But the manuscript definitely needs some serious work to be done before publication in AMT.

In this review I feel it justified to disclose my name, as I think there will be no harm to the overall review if the authors know, who I am. This will also likely support my arguments in the specific comments, as I do not intend to promote publication of my own papers but rather would like to draw the authors' attention to a couple of recent studies relevant to this work.

Specific comments:

title: The authors should make clear that the method is designed for dust aerosol and not for all kind of aerosols.

All: I would ask the authors to provide equation numbers in the revised manuscript.
p. 2 l.12ff: Normally I do not ask authors to cite my papers during review. But this case is specific, so I will deviate from the normal approach. The authors cite our first paper as well as the first paper in a full series describing the French LMD dust retrieval algorithm and infer the claim that satellite methods tend to underestimate effective radius. I do fully believe in the fact. But - there has been a lot of work done since the publication of these papers. For example by Klüser et al. (2015) in Remote Sensing of Environment and by Capelle et al. (2014) in ACP to name only the latest ones for the two methods referred to by the authors. In the Klüser et al. (2015) paper the authors also would see the impact of a variety of dust property assumptions on the particle size retrieval. The most important impact is the one of assumed particle sphericity. And here is also lies one of my biggest problem with the study: the authors do not at all acknowledge that particle shape has an extremely important impact on the retrieved particle size (for non-spherical particles also the definition of what effective particle size is is important!). We have done a small experiment with using different refractive indices and different assumed particle shapes for dust spectra and compared it to laboratory measurements and to Mie calculations. The results have been published as Klüser et al. in Journal of Quantitative Spectroscopy and Radiative Transfer this year. The authors might wish to look into that study (or a similar experiment by Legrand et al., published in 2014 in the same journal) for good descriptions of the impact of particle shape and dust composition on infrared extinction.

p. 2 l. 26: Satellite instruments measure radiance, not radiative flux. As flux is the radiation emitted into the hemisphere, there will be no way of measuring flux by satellite. Indeed for climate studies flux has to be estimated from satellite radiance, which is extremely difficulty due to the anisotropic nature of earth-leaving radiation.

p. 2 l.20ff: Here again I would strongly recommend to comment on the non-sphericity of dust particles and its impact on infrared radiation. Nevertheless the authors may go on with the Mie calculations. Indeed for broadband instruments such as SEVIRI the impacts might be small compared to other aspects, so the study doesn't loose its value

from describing the problem of non-sphericity. The authors could even prove this by a small comparison of Mie extinction efficiencies and, for example, T-matrix extinction efficiencies integrated over the relevant SEVIRI bands.

p. 3 section 3: It is common knowledge how to derive Mie extinction efficiency. So it is sufficient to present the Q_ext formula and provide any textbook on radiative transfer as reference. In the given way it might be worth to at least state the definition of x.

p. 3 l.23: Delete the sentence starting with "Berg et al. (2011)" as the large particle limiting case is of no interest for this study.

p. 4 l. 6: Be again aware of claiming proportionality which is not true (such as the one between Qe and I_0 and between I_0 and T). For example: if the claimed proportionality of 1/I_0 and Q_e would be true, that would signify, that for sufficiently large particles, regardless of AOD and temperature, the intensity would always be half of I_0 as Qe tends to be 2 for large particles. The authors will easily acknowledge that this cannot be true.

p. 4 l. 11: As already stated in the general comments the Ryleigh distribution falls from blue sky. Is there any physical motivation of using it? Otherwise it would be worth to quantitively prove that it is appropriate, for example with a small correlation experiment.

p. 4 all: As I have outlined above, the claims made here by the authors are overly simplified and in the context of radiative tranfer just wrong. The authors may have good reasons for sticking with these simplifications, but in that case they should spend a lt more effort in explaining. Also I am missing a description of the impact of dust layer height and temperature to the brightness temperature difference. These impacts are quite well understood and described in many papers.

p.5 l. 6: I am missing a description how these numerical values have been obtained.

p. 5 . l. 9: I do not understand this sentence. Does it mean, a LUT of the scaled brightness temperature difference has been created? For which d values? has the

surface emissivity been kept constant?

p. 5 l. 14f.: This is true only for intense dust storm conditions. Typically the majority of the radiation reaching the satellite still is emitted from the surface and has undergone no scattering at all. For example for a moderately thick dust optcial depth of 0.5 the transmittance is about 60% and for AOD of 1 the transmittance stil is approximately 37%. Taking into account emission by the dust layer itself (which the authors fail to comment on at all) these numbers reduce, depending on the single scattering albedo, but not in a way that the claim of the authors would become generally true. It is Acknowledging that these numbers refer to infrared optical depth and that the AOD ratio between the 11$\mu$m band and 0.55$\mu$m is somewhere around 2.7, these figures would translate to visible optical depths of 1.35 and 2.7, respectively.

p.5 l. 19: This dust flagging approach is by far too simple, see for example Ashpole et al. (2011) in JGR.

p.5 l21-29: I do not believe the claim that AOD should have hardly any impact of the brightness temperature, especially as almost every other study published in this field claims the opposite. If the authors wish to maintain this claim, they need to prove it by rigorous radiative transfer simulations.

p. 7 l. 11: One can compare model simulated particle sizes and satellite retrievals, but I would be extremely careful with calling this "verification of model results". Especially the proposed methods comes with so many simplifications and assumptions, that no modeller would believe it is more accurate than the modelled values.

p. 7 l.13-17: If this would be the aim of the aouthors, they would need a good AOD retrieval as well, see comments to Q_e-I_0 proportionality above.

p. 7 l. 21: This is the first time the words volcanic ash appear. The authors would need to explain why they are confident their method also works for volcanic ash (which in many ways is much more complicated than desert dust).

---

## Referee Comment (RC3) · Anonymous Referee #2 · 30 Sep 2016

This paper is focused on the interesting topic of dust cloud properties. The authors present a new remote sensing approach for retrieving the effective particle radius in dust clouds observed by SEVIRI. While the objective of the paper has merit, the methodology seems to be flawed or, at the very least, poorly explained. Therefore I do not recommend this paper for publication in its current form. Please see the comments below for specifics.

Section 1: • "The correlation between T and dust aerosols is rather complex and linked to many parameters. It is mainly caused by Aerosols Optical Depth(AOD), dust particle size and shape and the emissivity which in turn linked to dust chemical composition (e.g. Brindley et al. 2012; KluÌ́Lser et al. 2011)." The satellite measurements are also sensitive to the surface temperature, surface emissivity, atmospheric water vapor and temperature, and viewing angle. For optically thin dust clouds, the non-dust cloud

property components are especially relevant. Thus, I do not agree with the statement as written.

Section 2: • The authors should acknowledge that dust particles are not spherical. While I believe that the assumption of spherical particles is a secondary issue, motivation for treating dust particles as spheres should be provided. • It is also not clear as to what kind of size distribution was used in the Mie calculations. If the calculations were done for a single particle then the results are not at all representative of the particle size distributions present in nature. Also, the Mie calculations are a function of wavelength. Did the calculations take into account the SEVIRI spectral response functions?

Section 3: • The proportionality arguments do not make physical sense. The extinction efficiency is solely a function of the microphysical properties of the dust cloud, and is intrinsically independent of the incident radiation. In addition, the measured brightness temperature and incident radiation have a complex, non-linear, relationship. Further, the 8.7-12 um BTD is a complicated function of many variables and is not simply proportional to the 8.7 um surface emissivity. As such, the algorithm theoretical basis seems to be badly flawed, which is a primary reason I cannot recommend this paper for publication at this time. The authors need to provide a much more convincing argument for the theoretical basis. The generation of the various empirical relationships is also poorly explained. • The term "reemitted" is used. I recommend not using this term as matter emits radiation because it has a temperature. Once a photon is absorbed it should be considered dead and gone. • Even though the algorithm is restricted to pixels that meet certain BTD requirements thought to be related to optical depth the background atmosphere and surface and viewing angle will still influence the retrieval to varying degrees. The authors should include a sensitivity analysis that justifies their assumptions, as most modern retrieval methods do not make such assumptions.

---

## Author Comment (AC1) · 26 Oct 2016

We thank Reviewer# 3 for his comments. This reply is structured by introducing sections of the reviewer comments (in Italics) followed by a response. The page and line numbers of the updated version of the paper are used in the responses unless otherwise stated. The amended manuscript is attached in the supplement file.

(i) *The radiative transfer theory used here is far too simplistic. In reality, in the thermal infrared, one needs to consider both extinction (scattering and absorption) by, and emission from, the dust layer. Moreover, one has to consider emission from the sur-face and the underlying (and possibly overlying) atmosphere depen-*

dent on the dust height (atmospheric temperature structure) and the wavelength considered if one is to correctly interpret the satellite signal.

(ii) *There seems to be no appreciation of the Beer-Lambert law, or alternatively the differential nature of extinction, as radiation passes through a medium. Just considering absorption and removal by scattering by the dust layer alone would lead to an exponential dependence of the final 'intensity' on optical depth, which is itself a function of the extinction cross section. Add in emission, plus scattering into the upward direction and you will obtain the full radiative transfer equation (usually expressed in radiance although conversion to irradiance is possible if done properly).*

*In summary it should be noted that this reviewer is unconvinced that, even using the correct radiative transfer theory, and employing the simplifying assumptions that dust particles are spheres and have the same composition everywhere, there is enough independent information in a single BTD to extract size information. To do this I would suggest that at the very least, dust optical depth and height need to be known, and even then one would still have to account for the impact of confounding influences such as variable surface temperature, surface emissivity and water vapour content. It may be that there are 'regimes' of behaviour (e.g. dust plumes above a certain optical thickness) where size information can be extracted but I suggest the authors perform a much more comprehensive suite of (correct) radiative transfer calculations (explicitly simulating SEVIRI BTDs, including the relevant instrument characteristics) to look at whether what they are attempting to do is actually feasible. If they believe it is then they also need to come up with a much more convincing strategy for validating their results, including a traceable uncertainty analysis.*

The authors agree with the reviewer on the complexity of the correlation between brightness temperature and various dust layer properties and ground emissivity. An

acknowledgment of the complexity has been briefly introduced in the Introduction. Many studies tackled this problem through theoretical analysis but had limited success achieved to date in filling the gap between the observed and the modeled particle size. The reason, as the reviewer points out, is the high number of dependent variables that link the remotely sensed radiance and particle size in the radiative transfer theory. In addition, the high uncertainty in the in-situ estimation of these dependent variables which is an important source of the retrieval error (e.g. Merchant et al. 2006). The results of previous models seem to inherit the noise introduced by the vague estimation.

This paper approach tries to avoid this problem by exploiting the strong and dominate exponential effect of the particle size on the value of 8.7 and 12.0 $\mu$m Brightness Temperature Difference ($\Delta T_{8-12}$). Here we try to present this empirical evidence then use it to build a formula based mainly on observations and simplified conceptual model. However, we agree that basic support for the conceptual model through mathematical analysis cannot be avoided. The amended manuscript has corrected the over-simplified radiative transfer theory. Section 2 and 3 are substantially changed towards more theoretical bases that support the use of the empirical model. In a single thermal SEVIRI band, the effect of dust diameter is potentially "diluted" and difficult to see. In this paper, we show empirically that the change in effective diameter has very strong influence on the Brightness Temperature Difference (BTD) of 8.7 and 12.0 $\mu$m ($\Delta T_{8-12}$) over a surface of constant emissivity. In addition to the AbuDhabi case – (Figure 4), newly introduced Figures 5 & 6 add additional clarification to the exponential relation between $\Delta T_{8-12}$ and the effective dust diameter.

**Action:** more detailed mathematical description has been added to describe the basis that supports the model. Sections 2 and Section 3 have been rewritten. New Figures 5 and 6 have been introduced to explain the empirical evidence for the relationship between $\Delta T_{8-12}$ and effective diameter. More detailed description of the effect of surface emissivity, water vapour, and dust layer emissivity, height, non-sphericity has been included in section 3 and section 4.3 (Discussion of Results).

(iii) *The various relationships given near the start of section 3 are hence not correct. In fact, even if the earlier assumptions were ok I cannot see how they would logically follow. Why should the extinction efficiency be inversely proportional to the radiation incident on the dust layer? The former is an intrinsic property of the dust and is only dependent on the size distribution, shape of particles and composition. Similarly, brightness temperature is not directly proportional to the radiation incident on the dust. It is not even directly proportional to the intensity (as defined here) on the satellite radiometer but rather results from a non-linear conversion of the incident radiance using the Planck function.*

There is a typing mistake in that line. Instead of $Q_{ext} \; \alpha \; \frac{1}{I_0}$ , it should be $Q_{ext} \; \alpha \; 1 - \frac{I}{I_0}$ (*or* $Q_{ext} \; \alpha \; \frac{P_{removed}}{I_0}$) where $I$ is the radiance received by satellite radiometer. The pretext and the context that follows the relationships fits this intention. However, the authors agree that there is over simplification in the wording of that paragraph which resulted from using the Rayleigh-Jeans law which is not appropriate in thermal infrared part of the spectrum.

**Action:** A more detailed mathematical description using Plank's function has been presented in section 2 to justify the use of $\Delta Q_{ext} \; = \; 0$ with $\Delta T_{12-10}$ in estimating special cases of effective diameter $d$ .

(iv) *As noted above, aerosol optical properties are related to composition, shape, and size distribution. The use of Mie theory as given implicitly assumes that the particles are spherical which is rather unlikely for dust. Moreover, the authors simply use one set of refractive indices yet compare a number of different cases,*

*including African and Arabian dust events. One might anticipate significantly different compositions dependent on source. While the assumption of sphericity is likely to be less severe in the IR than the visible, at the very least some sort of sensitivity analysis should be performed to assess the impact of uncertainty in the dust composition on the resulting BTDs.*

The authors acknowledge that the variation of dust particle shape and chemical composition leads to a variation of the refractive index with a subsequent contribution to the total error. However, estimation of the error from non-sphericity and variation in chemical composition is a complex task and out the scope of this study. This is partly because it is still difficult to implement the available methods to quantify the effect of non-sphericity in estimating the extinction coefficient at a global scale. However, based on the performance of the other techniques that estimate the effective dust particle size, the method proposed within the paper is still potentially very useful for many applications with its current outcome.

**Action:** More detailed discussion about the limitation of the model including non-sphericity has been added in Section 4.3(Discussion of results), Page18.

(v) *As written, it is difficult to see whether the authors have any concept of the effect of a size distribution. Their Mie calculations appear to have been carried out for single particles (although I am not sure of this as the 'ringing' that one might expect to see in this case is absent). In reality, these responses will be weighted by the fraction of particles within each size bin, which will vary from dust event to dust event (and even during an individual dust event). Hence, when looking at real signals, the shapes in figure 1 will effectively be distorted differently for different distributions of particles such that fitting one empirical model is unlikely to be representative.*

MiePlot software (Laven, 2016) gives a choice to calculate a Mie solution for a range of particle size distributions. Here the particle sizes are assumed to be lognormally distributed in the range of [0.02 to 60 $\mu$m] although it is acknowledged that real distribution could be different. The selection of this range is based on the Ryder et al. (2013a, 2013b) report of volume distributions peaks between [10 to 60] m in fresh, heavy dust events which is the focus of interest for this calculation.

**Action:** This clarification has been added in section 2, Page4

(vi) *Similarly, have the authors taken the spectral width of the SEVIRI channels into account? It is not clear from what has been written. Since the filters are quite wide they will also affect the size of the signal seen and its variability. The viewing angle of the satellite will also affect the signals seen due to differential absorption through the atmosphere.*

The authors acknowledge that SEVIRI has wide spectral bands. On the one hand, the relatively wide range of SEVIRI spectral bands makes the signal less sensitive to using a Mie theory approximation of spherical shape compared with higher spectral resolution instruments onboard polar orbiting satellites. In addition, the authors acknowledge the effect of large view angle in the use of SEVIRI. On the other hand, there are advantages for operational use of SEVIRI in having a high temporal resolution product for dust particle size even if the there is a potential sacrifice in accuracy. Future study will involve testing the algorithm with higher spectral bands from VIIRS.

**Action:** An acknowledgement has been added in Page2 line 27.

(vii) *In the derivation of their model the authors appear to make the assumption that*

*the dust plume emissivity is the same as the surface emissivity (at least this is how it reads to this reviewer). This is not valid as, even if the composition is the same, the lofted particles are likely to be smaller and less densely packed than those at the surface.*

The authors acknowledge the difference in the emissivity between the ground and dust layer. However, it is found that using $\in_{8.7}^{2}$ gives more accuracy in the empirical model. The formula has been changed in the amended transcript to obtain better results, but it still shows the distinctive exponential pattern as described by Figures 5, 6 and 7.

**Action:** The model has been amended in the current version of the manuscript because it was found that by using the emissivity difference with other changes in the equation gave more accurate results.

(viii) *It is totally unclear where the 'measurements' at 15 micron used to fit the model have come from.*

Figure 1 shows two distinctive occasions when $Q_{ext10} - Q_{ext12} = 0$ for a dust layer. They correspond to effective diameter $d$ of 11.3 $\mu$m and 18.0 $\mu$m. In between the two values, 11.3$\mu$m and 18.0 $\mu$m, $Q_{ext12} - Q_{ext10} > 0$, and hence, as shown in section 2, $T_{12} - 0.991251\,T_{10} < 0$. The lowest value of $Q_{ext12} - Q_{ext10}$ is around 15 microns. The process was to look for a severe dust storm case where this condition is valid and take the corresponding $\Delta T_{8-12}$ at the same point of time and space.

**Action:** Further clarification has been added in section 2 to explain how to estimate $d$ from the corresponding brightness temperature when $Q_{ext12} - Q_{ext10} = 0$ in a severe dust case.

(ix) *In any case, using two clustered points to perform a curve fit such as that shown in figure 7 is, in my opinion, very bad science. I could fit any line I wanted through those points.*

The graph is a numerical solution for the coefficient of a known formula and a known line pattern which is already identified. It was not fitted from scratch. Excel solver was used to establishing the coefficients. The algorithm is based on searching for coefficients that correspond to a minimum square deviation from the model of the 12 sample points used in the solution. Using a known formula gives less freedom to fit an arbitrary line and still have a unique pattern. However, we agree that with more observations the model will become more precise.

**Action:** Two additional dust cases were added to the graph (Figure 11)

---

## Author Comment (AC3) · 26 Oct 2016

We thank Dr. Lars Klüser for his constructive and valuable comments. This reply is structurted by introducing sections of the comments (in Italics) followed by our response. The page and line numbers of the updated version of the paper are used in the responses unless otherwise stated. The amended manuscript is attached in the supplement PDF file.

*Specific comments: title: The authors should make clear that the method is designed for dust aerosol and not for all kind of aerosols. All: I would ask the authors to provide equation numbers in the revised manuscript.*

[Figure]

The authors agree with the suggested title as it reflect the content more clearly. Equation numbers have become necessary with the new amendment, which has a greater number of mathematical equations.

**Action:** The title has been changed to "Retrieval of effective dust diameter from satellite observations". Equation numbers have been provided.

*p. 2 l.12ff: Normally I do not ask authors to cite my papers during review. But this case is specific, so I will deviate from the normal approach. The authors cite our first paper as well as the first paper in a full series describing the French LMD dust retrieval algorithm and infer the claim that satellite methods tend to underestimate effective radius. I do fully believe in the fact. But - there has been a lot of work done since the publication of these papers. For example by Klüser et al. (2015) in Remote Sensing of Environment and by Capelle et al. (2014) in ACP to name only the latest ones for the two methods referred to by the authors. In the Klüser et al. (2015) paper the authors also would see the impact of a variety of dust property assumptions on the particle size retrieval. The most important impact is the one of assumed particle sphericity. And here is also lies one of my biggest problem with the study: the authors do not at all acknowledge that particle shape has an extremely important impact on the retrieved particle size (for non-spherical particles also the definition of what effective particle size is is important!). We have done a small experiment with using different refractive indices and different assumed particle shapes for dust spectra and compared it to laboratory measurements and to Mie calculations. The results have been published as Klüser et al. in Journal of Quantitative Spectroscopy and Radiative Transfer this year. The authors might wish to look into that study (or a similar experiment by Legrand et al., published in 2014 in the same journal) for good descriptions of the impact of particle shape and dust composition on infrared extinction.*

[Figure]

*p. 2 l.20ff.: Here again I would strongly recommend to comment on the non-sphericity of dust particles and its impact on infrared radiation. Nevertheless the authors may go on with the Mie calculations. Indeed for broadband instruments such as SEVIRI the impacts might be small compared to other aspects, so the study doesn't loose its value from describing the problem of non-sphericity. The authors could even prove this by a small comparison of Mie extinction efficiencies and, for example, T-matrix extinction efficiencies integrated over the relevant SEVIRI bands.*

The recent improvements in particle size retrieval have been revised and referenced in the revised manuscript. The authors acknowledge the complexity of the retrieving effective dust particle size using analytical approach. Apart from the recent improvements, many studies tackled this problem through theoretical analysis, but had limited success in filling the gap between the observed and the modelled particle size. The reason, as the reviewer points out, is the high number of dependent variables that link the remotely sensed radiance and particle size in the radiative transfer theory. The high uncertainty in the approximation of variables such as the chemical composition and particle shape, might be one reason to limit the advance in improving the accuracy of retrievals. The results of previous particle size models seem to inherit the noise introduced by the vague estimation of the many dependent variables. In a single thermal SEVIRI band, the effect of dust diameter is potentially "diluted" and difficult to see while the case turn out to be different in Brightness Temperature Difference of 8.7 and 12.0$\mu$m ($\Delta T_{8-12}$). This study approach aim to avoid the inherited noise of many dependent variables by exploiting the strong and dominate exponential effect of the particle size on the value of $\Delta T_{8-12}$. Here we try to present the empirical evidence of this relation and then use it to build a formula based mainly on empirical data and a simplified conceptual model.

**Action:** To make the thesis of the paper clearer, Section 3 has been rewritten to include more details to explain the empirical evidence of dominate exponential effect of the particle size on the value of $\Delta T_{8-12}$. More detailed description of the effect

of surface emissivity, water vapour, and dust layer properties has been also included in section 3. In section 4.3(Discussion of Results) more details discussion on the limitation of the model has been presented.

*p. 2 l. 26: Satellite instruments measure radiance, not radiative flux. As flux is the radiation emitted into the hemisphere, there will be no way of measuring flux by satellite. Indeed for climate studies flux has to be estimated from satellite radiance, which is extremely difficulty due to the anisotropic nature of earth-leaving radiation.*

**Action:** corrected

*p. 3 section 3: It is common knowledge how to derive Mie extinction efficiency. So it is sufficient to present the Q_ext formula and provide any textbook on radiative transfer as reference. In the given way it might be worth to at least state the definition of x.*

The aim of that introduction is to explain the relation between the extinction efficiency difference, effective particle diameter and BTD.

**Action:** Some common-knowledge details have been removed in Section2 Page3.

*p. 3 l.23: Delete the sentence starting with "Berg et al. (2011)" as the large particle limiting case is of no interest for this study.*

**Action:** done.

*p. 4 l. 6: Be again aware of claiming proportionality which is not true (such as*

*the one between Qe and I_0 and between I_0 and T). For example: if the claimed proportional-ity of 1/I_0 and Q_e would be true, that would signify, that for sufficiently large particles, regardless of AOD and temperature, the intensity would always be half of I_0 as Qe tends to be 2 for large particles. The authors will easily acknowledge that this cannot be true.*

*p. 4 all: As I have outlined above, the claims made here by the authors are overly simplified and in the context of radiative tranfer just wrong. The authors may have good reasons for sticking with these simplifications, but in that case they should spend a lt more effort in explaining.*

There is a typing mistake in the referenced line number. Instead of $Q_{ext}\ \alpha\ \frac{1}{I_0}$ , it should be $Q_{ext}\ \alpha\ (1 - \frac{I_r}{I_0})\ (from\ Q_{ext} = \frac{P_{removed}}{I_0})$ where ($I_r$) is the radiance received by satellite radiometer. The pretext in the introduction and the context that follows the relationships fits this intention. However, the Authors agree that there is over simplification in the analogy of the radiative transfer on *page 4* in the referenced manuscript because of using Rayleigh-Jeans law, which is not appropriate in thermal infrared part of the spectrum.

**Action:** more detailed mathematical description using Plank's function to justify using the difference of extinction factor has been presented.

*. . . Also I am missing a description of the impact of dust layer height and temperature to the brightness temperature difference. These impacts are quite well understood and described in many papers.*

It is clear that the dust layer brightness temperature decreases with height mainly due to decreasing ambient temperature. But the change of BTD with height is less obvious. Brindley & Russell (2006) and Merchant et al. (2006) used radiative transfer models to show that $\Delta T_{8-10}$ changes with changing the dust layer height and AOD, extinction

coefficient and absorption (Emissivity). Taking into the account that AOD, extinction coefficient and emissivity are all a function of the particle size, the change in $\Delta T_{8-10}$ also conveys information of the particle size. Hence, the change in $\Delta T_{8-10}$ value can be partly attributed to change in the particle size and it is misleading to conclude, based on these studies, that there will be a big impact on the accuracy of the effective particle size retrieval using $\Delta T_{8-12}$.

**Action:** The above description of the dust layer height and temperature change has been added to Page 7, Line 19.

*p. 4 l. 11: As already stated in the general comments the Ryleigh distribution falls from blue sky. Is there any physical motivation of using it? Otherwise it would be worth to quantitively prove that it is appropriate, for example with a small correlation experiment.*

The main motivation derive is from the empirical observations which have been more clearly explained in section 3 of the revised manuscript. The formula has been changed to obtain better results but it still shows the distinctive exponential pattern as described by Figures 5, 6 and 7.

*p.5 l. 6: I am missing a description how these numerical values have been obtained.*

Excel solver (Fylstra et al., 1998; Harris, 1998) was used with the values of $d$ and the corresponding $\Delta T_{8-12}$ from the four dust cases described above to solve for the $\frac{\Delta T_{8-12}}{(E+\Delta\varepsilon)} = (\frac{A\ d^3}{(e^{\alpha d}-1)} - C)$ model coefficients. The technique was based on converging the solutions of the points towards a minimum sum of square deviation.

**Action:** This detail has been added in Page 14, line 1.

[Figure]

*p. 5 . l. 9: I do not understand this sentence. Does it mean, a LUT of the scaled brightness temperature difference has been created? For which d values? has the surface emissivity been kept constant?*

In the amended paper (Page14, Line5), the LUT was be created for $d$ versus $\frac{\Delta T_{8\,-12}}{(E+\Delta\varepsilon)}$ . Neither $\Delta T_{8\,-12}$ nor emissivity are constants. But, since $\frac{\Delta T_{8\,-12}}{(E+\Delta\varepsilon)}$ is a function of $d$, there is a unique value of $\frac{\Delta\,T_{8\,-12}}{(E+\Delta\varepsilon)}$ corresponding to every value of $d$.

*p. 5 l. 14f.: This is true only for intense dust storm conditions. Typically the majority of the radiation reaching the satellite still is emitted from the surface and has undergone no scattering at all. For example for a moderately thick dust optcial depth of 0.5 the transmittance is about 60% and for AOD of 1 the transmittance stil is approximately 37%. Taking into account emission by the dust layer itself (which the authors fail to comment on at all) these numbers reduce, depending on the single scattering albedo, but not in a way that the claim of the authors would become generally true. It is Ac-knowledging that these numbers refer to infrared optical depth and that the AOD ratio between the 11$\mu$m band and 0.55$\mu$m is somewhere around 2.7, these figures would translate to visible optical depths of 1.35 and 2.7, respectively.*

*p.5 l. 19: This dust flagging approach is by far too simple, see for example Ashpole et al. (2011) in JGR.*

We agree that using the words "dominating" and "most of" are not appropriate since we are talking about a single band behaviours of 10.8 and 12.0 $\mu$m bands, not the difference between the two. We agree with the reviewer's explanation which does not contradict the idea intended to be transmitted to the reader in that paragraph.

**Action:** The paragraph starting at Page 19, Line 7 has been rephrased.

*p.5 l21-29: I do not believe the claim that AOD should have hardly any impact of the brightness temperature, especially as almost every other study published in this field claims the opposite. If the authors wish to maintain this claim, they need to prove it by rigorous radiative transfer simulations.*

We are not claiming that AOD does not affect brightness temperature. To the contrary, we are saying that AOD is strongly related to the apparent brightness temperature. Rather our claim is BTD ($\Delta T_{8 - 12}$) has limited correlation with AOD as shown by the presented case (Page 9, Line 15).

**Action:** The paragraph (Page 9, Line 15) has been rephrased to make the case clearer.

*p. 7 l. 11: One can compare model simulated particle sizes and satellite re-trievals, but I would be extremely careful with calling this "verification of model results". Especially the proposed methods come with so many simplifications and assumptions, that no modeller would believe it is more accurate than the modelled values.*

**Action:** The phrase (Page 20, Line 3) has been changed to "To provide an independent reference data for atmospheric aerosols model comparison".

*p. 7 l.13-17: If this would be the aim of the authors, they would need a good AOD retrieval as well, see comments to Q_e-I_0 proportionality above.*

We agree that the performance of the solar power systems depends on the turbidity of the atmosphere; this depends on AOD, as well as the number of dust particle pre-cipitates over the solar panels which correlate with the effective particle diameter. The

technique proposed in the paper can give an indication of the amount of dust that will precipitate on solar energy systems from an upcoming dust event.

**Action:** The statement has been changed accordingly (Page 20, line 11).

*p. 7 l. 21: This is the first time the words volcanic ash appear. The authors would need to explain why they are confident their method also works for volcanic ash (which in many ways is much more complicated than desert dust).*

**Action:** Volcanic ash has been removed

**Supplement:**

**Retrieval of effective dust diameter from satellite observations**

Humaid Al Badi1,3, John Boland1, David Bruce2

1Centre for Industrial and Applied Mathematics, University of South Australia, Mawson Lakes, 5095, Australia.

2Natural and Built Environments Research Centre, University of South Australia, Mawson Lakes, 5095, Australia.

3 Directorate General of Meteorology, Public Authority for Civil Aviation, Oman.

Correspondence to: Humaid Al Badi (h.albadi@gmail.com)

Abstract. Dust aerosol particle size plays a crucial role in determining dust cycle in the atmosphere and the extent of its impact on the other atmospheric parameters. The in-situ measurements of dust particle size are very costly, spatially sparse and time-consuming. This paper presents an algorithm to retrieve effective dust diameter using infrared band Brightness Temperature Difference (BTD) from SEVIRI (the Spinning Enhanced Visible and InfaRed Imager) on the Meteosat satellite. An empirical model was constructed that directly relates BTD of 8.7 and 12.0  $\mu$ m bands ( $\Delta T_{8-12}$ ) to dust effective diameter. Three case studies are used to test the model. The results showed consistency between the model and in-situ aircraft measurements. A severe dust storm over the Middle-East is presented to demonstrate the use of the model. This algorithm is

15 expected to contribute to filling the gap created by the discrepancies between the current size distributions retrieval techniques and aircraft measurements. Potential applications include enhancing the accuracy of atmospheric modelling and forecasting horizontal visibility as well as solar energy system performance over regions affected by dust storms.

**1. Introduction**

5

- 20 Aerosols including dust have a significant impact on the climate through a range of complex mechanisms. In the short term, the effects of aerosol variability generate perturbations in atmospheric turbidity. Aerosols alter atmospheric turbidity by modifying the short-wave solar radiation and terrestrial long wave radiation through scattering and absorption. The amount of absorption and forward and backward scattering depends on the concentration, size distribution and chemical composition of aerosol particles. Despite continued research, the multiple dust aerosol effects are still poorly represented in climate
- 25 models which lead to substantial uncertainty (Ben-Ami *et al.*, 2010; Boucher *et al.*, 2013). One of the reasons for such uncertainty might be the scarcity of adequate and routine measurements of dust aerosol properties. Most of the particle size in-situ measurements in operation take place on the ground by sampling the precipitated dust aerosols (Afeti and Resch, 2000; Sunnu, Afeti and Resch, 2008). Ground measurements are not sufficient because aerosols have different and dynamic vertical distributions. Limited aircraft campaigns have been conducted to sample the aerosol particle size in the atmosphere (
- 30 e.g. Tanré et al. 2003; Müller et al. 2012; Ryder et al. 2013b; Ryder et al. 2013a).

The sparsity of in-situ dust sampling opened the door for remote sensing techniques to fill the gap. Nakajima et al. (1996) and Dubovik & King (2000) developed inversion algorithms for retrieval of the aerosol volume distribution  $\left(\frac{dV}{d\ln r}\right)$  from Sun and sky radiance ground measurements, such as AErosol RObotic NETwork (AERONET). From space, Atmospheric Infrared Sounder (AIRS) observations made at 9.4 µm are used to retrieve dust effective radius (Pierangelo et al. 2005).

5 Klüser et al. (2011) also used a Singular Vector Decomposition method on observed spectra from the Infrared Atmospheric Sounding Interferometer (IASI) to extract dust effective radius.

However, the current techniques do not satisfy the need for accurate dust particle size data. Discrepancies have been observed in retrieving the size distributions between the AERONET algorithm (Dubovik and King, 2000), the Sky Radiation 10 (SKYRAD) algorithm (Nakajima et al., 1996) and aircraft measurements (Estellés et al., 2012; Ryder et al., 2015). The comparison between in-situ sampling during Saharan Mineral Dust Experiment (SAMUM) 2006 and AERONET effective radii retrieval showed that the Sun photometer observations are smaller by a factor of approximately two compared with the in-situ observations (Müller *et al.*, 2012). The underestimation of the dust size distribution also appears to be a common pattern among the current satellite algorithms (Klüser et al., 2011; Pierangelo et al. 2005). The abundance of large dust 15 aerosol particles over desert surfaces might have a role in reducing the accuracy of the current sun-photometer and satellite techniques (Ryder et al. 2013b). Recently, signs of improvements have been demonstrated if a detailed analysis of particle shape and composition are considered in the retrieval models (Capelle et al., 2014; Legrand et al., 2014; Klüser et al., 2015, 2016). One problem that limits the advance in improving particle size retrieval is the noise introduced by the vague estimation of the many dependent variables. This study aims to avoid this problem by using the strong and dominate 20 exponential effect of the particle size on the value of 8.7 and 12.0  $\mu$ m Brightness Temperature Difference ( $\Delta T_{8-12}$ ). The empirical evidence for  $\Delta T_{8-12}$  and the dust particle size is presented and then used to develop a formula based mainly on observational data and a simplified conceptual model.

Infrared window bands 12.0, 10.8 and 8.7 µm from SEVIRI (the Spinning Enhanced Visible and InfaRed Imager) on 25 Meteosat satellite are widely used to track dust plumes in operational meteorological applications. In 2003 the first METEOSAT Second Generation Satellite was launched carrying SEVIRI. It has an unprecedented temporal resolution of 15 minutes over the Sahara Desert and the West Asia regions where the primary dust sources in the world are located. SEVIRI has relatively wide spectral bands compared with polar orbiting satellite imagers. On the one hand, the relatively wide range of SEVIRI spectral bands makes the signal less sensitive to using approximation such as Mie theory approximation of 30 spherical shape compared to higher spectral resolution instruments (Rees and Rees, 2013; Klüser et al., 2015). On the other

hand, there are other advantages for operational use of SEVIRI in having a high temporal resolution product for dust particle size even if the there is a potential sacrifice in accuracy.

The Dust Red, Green and Blue (Dust RGB) image composite corresponding to infrared window band combination of 12.0-10.8, 10.8-8.7 and 10.8 µm respectively, is one of the most used combinations to track dust clouds (Eumetsat-MSG, 2016). The Dust RGB composite uses the fact that the change of the band's brightness Temperature (T) is strongly correlated with the change in the scattering and absorption caused by dust variability in the atmosphere. That is, the dust aerosol variability

- 5 alters the radiance falling on the satellite radiometer from which T is calculated. The problem of retrieving dust particle size through analytical approach is complex because a dust layer alone has many variables which might affect a single band brightness temperature T even if the other variables such as surface temperature, surface emissivity, atmospheric water vapor and temperature, and viewing angle are known. A dust layer affects T mainly by Aerosol Optical Depth (AOD), dust particle size and shape and the dust aerosol emissivity which is in turn linked to dust chemical composition (e.g. Brindley et
- 10 al. 2012; Klüser et al. 2011). The uncertainty in the approximation of these many depended variables might be one reason to limit the advance in improving the accuracy of dust size retrieval through detailed analytical approach. The results of previous particle size models seem to inherit the noise introduced by the vague estimation of the dependent variables. In a single thermal SEVIRI band, the effect of dust diameter is potentially "diluted" and difficult to see while the case is different with  $\Delta T_{8-12}$  as it is presented later. This study aim to avoid the inherited noise of many dependent variables by exploiting
- 15 the strong and dominate exponential effect of the particle size on the value of  $\Delta T_{8-12}$ . Here the empirical evidence of this relation is presented and then use it to build a model based mainly on empirical data and a simplified conceptual model. To build the empirical model that link  $\Delta T_{8-12}$  with effective diameter (d), a full range of  $\Delta T_{8-12}$  versus d observations were needed. That is, observation from light to severe dust storms. For relatively light dust emission, the effective diameter sampled by Fennec aircraft campaign during June 2011 over West Africa (Ryder et al., 2013) was used with the corresponding  $\Delta T_{8-12}$ . For severe dust storms, the Mie extinction efficiency factor is used to estimate the effective diameter
- 20

for  $\Delta Q_{ext} = 0$  which then used in building model.

**2. Estimating brightness temperatures that correspond to $\Delta Q_{ext} = 0$**

25

To assess how much a spherical dust particle scatters light, the extinction efficiency factor  $Q_{ext}$  needs to be introduced. Any particle of diameter d which intersects the radiation path will remove power from the incident radiation with intensity  $L_0$  by an amount  $P_{removed}$

$$P_{removed} = C_{ext} L_0 \tag{1}$$

where  $C_{ext}$  is the extinction cross section (Hahn, 2009).

The extinction efficiency factor  $Q_{ext}$  for a spherical particle  $Q_{ext}$  is given by Mie's solution for Maxwell's electromagnetism equations as

$$Q_{ext} = \frac{C_{ext}}{C_{geo}} = \frac{2}{x^2} \sum_{n=1}^{\infty} (2n+1)Re\{a_n(x,m,\Psi_n,\xi_n) + b_n(x,m,\Psi_n,\xi_n)\}$$
(2)

where x is called the size parameter and equals  $\frac{\pi d}{\lambda}$ ,  $C_{geo}$  is the geometrical cross section and equals  $\frac{\pi d^2}{4}$ , m is the refractive index,  $a_n(x, m, \Psi_n, \xi_n)$  and  $b_n(x, m, \Psi_n, \xi_n)$  are Mie scattering coefficients derived from solving Maxwell's equations, and  $\Psi$  and  $\xi$  are the Ricatti-Bessel functions (Hahn, 2009).

Eq(2) is solved numerically for any given x and m. The refractive index is wavelength and chemical composition dependent.

5 In this paper, the Di Biagio et al. (2014) estimation of the refractive index *m* has been used where,  $m_{8.7} = 1.10 + 0.20i$ ,  $m_{10.8} = 1.9 + 0.25i$ ,  $m_{12.0} = 1.75 + 0.40i$  are the values given for the 8.7, 10.8 and 12.0 µm respectively. These refractive index estimations were made in the laboratory for five dust samples collected during dust events originated from different Western Saharan and Sahelian areas (Di Biagio *et al.*, 2014). In this paper variation in chemical composition of dust particles from different sources has not been taken into account.

Figure 1: Extinction Efficiency Factor  $Q_{ext}$  at 8.7, 10.8 and 12.0  $\mu$ m wavelengths versus the particle diameter.

15

10

MiePlot software (Laven, 2016) gives a choice to calculate Mie solution for a range of particle size distributions. Here the particle sizes are assumed to be lognormally distributed in the range of [0.02 to 60  $\mu$ m] although it is acknowledged that real distribution could be different. The selection of this range is based on the Ryder et al. (2013a, 2013b) report of volume distribution peaks between [10 to 60]  $\mu$ m in fresh, heavy dust events which is the focus of interest for this calculation. Figure 1 shows the calculated Mie extinction efficiency factor  $Q_{ext}$  for particle diameter from 1 to 50  $\mu$ m. As Figure 1 shows, Mie

theory predicts a significant change in the thermal infrared 12.0, 10.8 and 8.7  $\mu$ m extinction efficiency factor when the particle diameter lies between 1 and 20  $\mu$ m. This dust range covers the reported effective dust particle size range during the Fennec 2011 aircraft dust sampling campaign over West Africa which was between 2.3 to 19.4  $\mu$ m for several dust events (Ryder et al. 2013b).

5

Since
$$C_{ext} = \frac{P_{remove}}{L_0} = \frac{L_0 - L_r}{L_0} = 1 - \frac{L_r}{L_0}$$
, hence

$$Q_{ext} = \frac{4}{\pi d^2} (1 - \frac{L_r}{L_0})$$
(3)

Where  $L_0$  is the surface radiance of a narrow spectral band,  $L_r$  is the radiance for the same narrow band received by a satellite radiometer, which is given by the integral of Plank's function for apparent brightness temperature T

$$\int_{\lambda_1}^{\lambda_2} L_{r\lambda} d\lambda = \frac{hc^3}{k\lambda^3 (e^{\frac{hc}{\lambda kT}} - 1)}$$
(4)

10 Following Widger Jr and Woodall, (1976); and Rees and Rees, (2013), for a finite range of wavelength

$$L_r = \pi \int_{\lambda_1}^{\lambda_2} L_\lambda d\lambda = \sigma T^4 (f(x_1) - f(x_2)) = \pi \int_{\lambda_1}^{\lambda_2} L_\lambda d\lambda = \sigma T^4 \Delta f$$
(5)

Where  $\sigma$  is the Stefan–Boltzmann constant,  $x_i = \frac{hc}{\lambda_i kT}$ ,  $\Delta f = f(x_2) - f(x_1) = \frac{15}{\pi^4} \int_0^{x_2} \frac{x^3 dx}{e^{x_{-1}}} - \frac{15}{\pi^4} \int_0^{x_1} \frac{x^3 dx}{e^{x_{-1}}}$

Substitute the corresponding value of  $L_r$  in Eq (5) into Eq (3) for the two SEVIRI bands 10.8 [9.8 to 11.8µm], 12.0 [11.0 to 13.0] µm and solve for the difference of the extinction efficiency factor  $Q_{ext}$  we get,

$$Q_{ext10} - Q_{ext12} = \frac{4}{\pi d^2} \left( \frac{L_{r12}}{L_{012}} - \frac{L_{r10}}{L_{010}} \right) = \frac{4}{\pi d^2} \left( \frac{\sigma T_{12}^4 \Delta f_{12}}{\varepsilon_{12} \sigma T_s^4 \Delta f_{s12}} - \frac{\sigma T_{10}^4 \Delta f_{10}}{\varepsilon_{10} \sigma T_s^4 \Delta f_{s10}} \right)$$
(6)

Where  $T_s$  is the surface temperature and  $\varepsilon_{\lambda}$  is the spectral surface emissivity.

15 when  $Q_{ext8} - Q_{ext12} = 0$

$$\frac{\sigma T_{12}^4 \Delta f_{12}}{\varepsilon_{12} \sigma T_s^4 \Delta f_{s12}} = \frac{\sigma T_{10}^4 \Delta f_{10}}{\varepsilon_{10} \sigma T_s^4 \Delta f_{s10}} \rightarrow \frac{T_{12}^4 \Delta f_{12}}{\varepsilon_{12} \Delta f_{s12}} = \frac{T_{10}^4 \Delta f_{10}}{\varepsilon_{10} \Delta f_{s10}}$$

Which gives

$$\frac{T_{12}}{T_{10}} = \left(\frac{\varepsilon_{12}\Delta f_{10}\Delta f_{s12}}{\varepsilon_{10}\Delta f_{s10}\Delta f_{12}}\right)^{\frac{1}{4}}$$
(7)

The mean emissivity values of barren surfaces at 10.8µm and 12.0µm bands are 0.9478 and 0.9659 respectively (Trigo *et al.*, 2008). For a typical severe dust storm over the Middle East, the temperature of a dust cloud drops to 275

Kelvin while the surface temperature during the day is 300 Kelvin on average. Adopting the numerical solution for f(x) given by (Rees and Rees, 2013), under such conditions and for a thick dust layer, Eq (7) implies:

$$Q_{ext10} - Q_{ext12} = 0 \quad \rightarrow T_{12} = 0.991251 T_{10} \tag{8}$$

Obviously, more precise results can be obtained if localised emissivity data of  $\varepsilon_{10}$  and  $\varepsilon_{12}$  are used. Figure 1 shows two distinctive occasions when  $Q_{ext10} - Q_{ext12} = 0$  for a dust layer. They correspond to effective diameter d, of 11.3 µm and 18.0 µm. In between these two values,  $Q_{ext12} - Q_{ext10} > 0$ , and hence  $T_{12} - 0.991251 T_{10} < 0$ . Using the condition in Eq (8) with a real data of a dust storm, it is possible to find the values of  $T_{12}$ ,  $T_{10}$  and  $T_8$  which correspond to the effective diameter d 11.3µm and 18.0 µm for a severe dust storm. The values will be used to solve for the coefficients of a generalised model in section 3.

**10 3. An empirical formula to link $\Delta T_{8-12}$ and effective diameter**

5

The measured Earth radiance at a satellite instrument has two components, a surface contribution, and an atmospheric contribution. The expression for of satellite remotely sensed spectral radiance  $L_{\lambda}$  can be simplified mathematically to the following form of the radiative transfer equation (Kerr *et al.*, 1992; Walton *et al.*, 1998; Dash *et al.*, 2002; Jin *et al.*, 2015):

$$L_{\lambda} = \varepsilon_{\lambda} t_{\lambda} B_{\lambda}(T_s) + (1 - t_{\lambda}) B_{\lambda}(T_a)$$
(9)

Where  $L_{\lambda}$  is spectral radiance received by a satellite instrument,  $\varepsilon_{\lambda}$  is surface emissivity,  $t_{\lambda}$  is the transmittance of the 15 atmosphere,  $B_{\lambda}(T_s)$  Planks function for surface temperature  $T_s$ ,  $B_{\lambda}(T_a)$  Plank's function for the average temperature of the 15 atmosphere. This form of radiative transfer equation assumes no downward radiance and that atmospheric transmittance 16 variance results mainly from different absorption coefficients and forward scattering. 17 The transmittance  $t_{\lambda}$  is given by:

$$t_{\lambda} = e^{-\iint \pi n(r) Q_{ext}(r,\lambda)r^2 \, dr ds} = e^{-\int \pi \sigma_{ext}(d,\lambda)ds} = e^{-\tau_{\lambda}} \qquad (\text{Ackerman, 1997})$$
(10)

Where n(r) is the aerosol size distribution of radius r,  $Q_{ext}$  is the extinction efficiency factor,  $\sigma_{ext}$  is the extinction 20 coefficient, d, is the particle diameter,  $\tau_{\lambda}$  is the AOD.

Split window thermal infrared brightness temperature is commonly used to retrieve land surface temperature e.g. (Kerr *et al.*, 1992; Sobrino *et al.*, 1994; Dash *et al.*, 2002; Jin *et al.*, 2015). For SEVIRI band 8.7 µm and 12.0 µm Eq (9) can be given as:

$$L_8 = \varepsilon_8 t_8 B_8(T_s) + (1 - t_8) B_8(T_a)$$
(11)

$$L_{12} = \varepsilon_{12} t_{12} B_{12}(T_s) + (1 - t_{12}) B_{12}(T_a)$$
(12)

The split-window equation is formed by substituting expanding Plank's Function  $B_{\lambda}(T)$  and solving the two equations for  $T_s$ (e.g. Sobrino and Raissouni, 2000; Dash *et al.*, 2002):

$$T_s = T_8 + \beta_1 (T_8 - T_{12}) + E_1 \tag{13}$$

Where  $\beta_1$  accounts for the atmospheric transmittance  $\beta_1 = \frac{1-t_8}{t_8-t_{12}}$  and  $E_1$  accounts for the emissivity from different sources (e.g. Sobrino and Romaguera, 2004). It is also possible to form the following equation by solving Eq (11) and Eq (12) for  $T_s$  using 8.7 µm and 10.8 µm bands

$$T_s = T_8 + \beta_2 (T_8 - T_{10}) + E_2 \tag{14}$$

Subtracting Eq (14) from Eq (13) results in:

$$0 = \beta_1 (T_8 - T_{12}) - \beta_2 (T_8 - T_{10}) + E_1 - E_2$$

For a dry air mass or if significant dust aerosols exist, the difference of 12.0  $\mu$ m and 10.8  $\mu$ m brightness temperatures is small compared to a much bigger difference between 12.0  $\mu$ m and 8.7  $\mu$ m (Eumetsat-MSG, 2016). Thus, it is assumed here that  $(T_8 - T_{10}) = (T_8 - T_{12})$ , which leads to:

$$\Delta T_{8-12} = \frac{E_2 - E_1}{\beta_1 - \beta_2} \tag{15}$$

Eq (15) presents  $\Delta T_{8-12}$  as a function of the difference of the emissivity coefficients to the difference of the transmissivity coefficients. E2 and E1 represent mainly surface emissivity, water vapour (Sobrino and Romaguera, 2004) and dust in case 10 of high dust concentration. All the three bands have equally high water emissivity with small differences between them. Since a difference is taken  $(E_2 - E_1)$ , the net contribution of water vapour emissivity in  $\Delta T_{8-12}$  value is expected to be low and have a minor contribution. Brindley & Russell (2006) experiments with radiative transfer model and SEVIRI showed that BTD  $\Delta T_{8-10}$  and  $\Delta T_{12-10}$  variability rang with water vapour variability is less than 0.2 Kelvin which is very low variation given that the  $\Delta T_{8-10}$  can get up to 15 Kelvin (Eumetsat-MSG, 2016). The same is also expected from dust layer 15 emissivity because the difference in the dust aerosol emissivity between the two bands is also small when the particle size is between 0.1 to 37 µm (Takashima & Masuda, 1987). However, there is a significant difference in the ground emissivity between the bands; 12.0 and 8.7 µm. Most of the Earth's surface has a near blackbody 12.0 µm emissivity of 0.93 and higher. Sandy desert surfaces usually have a much lower 8.7 µm emissivity, typically around 0.65. Thus, this significant difference is expected exhibit itself in  $\Delta T_{8-12}$  value. In this paper, the Global Infrared Land Surface Emissivity Database has been used (Seemann et al., 2008). Figure 2 shows the emissivity of 8.3 µm with strong variability while 12.1 µm emissivity 20 in Figure 3 is more homogenous around a relatively high value.

It is clear that the dust layer brightness temperature decrease with height due to mainly decreasing ambient temperature, but the change of thermal infrared BTD with height is less obvious. Brindley & Russell (2006) and Merchant et al. (2006) used radiative transfer models to show that  $\Delta T_{8-10}$  changes with changing the dust layer height and AOD, extinction coefficient and absorption (Emissivity). Taking into account that AOD, extinction coefficient and emissivity are all a function of the particle size, the change in  $\Delta T_{8-10}$  convey information of the particle size too. Hence, the change in  $\Delta T_{8-10}$  value can be attributed partly as a change in particle size and it is misleading to conclude, based on these studies, that there will be a big impact on the accuracy of the effective particle retrieval using  $\Delta T_{8-12}$ .

---

## Author Comment (AC4) · 26 Oct 2016

We thank Reviewer#2 for his comments. This reply is structured by introducing sections of the comments (in Italics) followed by a response. The page and line numbers of the updated version of the paper are used in the responses. The amended manuscript is attached in the supplement PDF file.

*Section 1: "The correlation between T and dust aerosols is rather complex and linked to many parameters. It is mainly caused by Aerosols Optical Depth(AOD), dust particle size and shape and the emissivity which in turn linked to dust chemical composition (e.g. Brindley et al. 2012; Kluser et al. 2011)." The satellite measurements are also sensitive to the surface temperature, surface emissivity, atmospheric water*

[Figure]

*vapor and temperature, and viewing angle. For optically thin dust clouds, the non-dust cloud property components are especially relevant. Thus, I do not agree with the statement as written.*

We agree with the reviewer that there are many other variables which affect brightness temperature. The statement was intended to highlight the complexity of correlation between $T$ and the dust optical properties. It did not intend to refer to all the variables in the radiation transfer equation. To ensure that we are writing about dust properties only, we started the sentence with "The correlation between $T$ and dust aerosols". The statement is trying to convey that even if the other variables are known, the problem of retrieving dust particle size through analytical approach is still complex because a dust layer alone has many variables which might affect the $T$.

**Action:** More detail has been added to the statement to ensure clarity of the paper - Page 3, line 4 to 15.

*Section 2: The authors should acknowledge that dust particles are not spherical. While I believe that the assumption of spherical particles is a secondary issue, motivation for treating dust particles as spheres should be provided.*

The authors acknowledge that the variation of dust particle shape and chemical composition leads to a variation of the refractive index with a subsequent contribution to the total error. However, estimation of the error from non-sphericity and variation in chemical composition is a complex task and out the scope of this study. This is partly because it is still difficult to implement the available methods to quantify the effect of non-sphericity in estimating the extinction coefficient developed by other researchers, e.g.(Cheng et al. 2010; Dubovik et al. 2002; Dubovik et al. 2002b; Wang et al. 2003). The chemical composition also has an effect on optical properties of dust aerosols. Different dust sources have different dust composition. As non-sphericity do, the chemical composition affects the refractive index of dust. Klüser et al.( 2015, 2016) give a

detailed analysis on the effect of non-sphericity and chemical composition on spectral bands in the thermal infrared region. At this stage, the extent of the effect of the non-sphericity and chemical composition is not known when taking the brightness temperature difference between two bands. However, it is still crucial to develop localized accuracy assessment of the algorithm to compensate the difference in the dust particle morphology and composition.

**Action:** This clarification has been added in Section 4.3 (Discussion of Results), Page18.

*It is also not clear as to what kind of size distribution was used in the Mie calculations. If the calculations were done for a single particle then the results are not at all representative of the particle size distributions present in nature.*

MiePlot software (Laven, 2016) gives a choice to calculate a Mie solution for a range of particle size distributions. Here the particle sizes are assumed to be lognormally distributed in the range of [0.02 to 60 $\mu$m] although it is acknowledged that real distributions could be different. The selection of this range is based on the Ryder et al. (2013a, 2013b) report of volume distributions peaks between [10 to 60] m in fresh, heavy dust events which is the focus of interest for this calculation.

**Action:** This clarification has been added in section 2, Page4.

*Also, the Mie calculations are a function of wavelength. Did the calculations take into account the SEVIRI spectral response functions?*

Yes, it has done separately for the 8.7, 10.8 and 12.0 $\mu$m as explained in Page 3 Line 25. The empirical formula is built for SEVIRI, hence, we believe the numerical constants in the formula will probably change if another instrument is used.

*Section 3: The proportionality arguments do not make physical sense. The extinction efficiency is solely a function of the microphysical properties of the dust cloud, and is intrinsically independent of the incident radiation. In addition, the measured brightness temperature and incident radiation have a complex, non-linear, relationship. Further, the 8.7-12 um BTD is a complicated function of many variables and is not simply proportional to the 8.7 um surface emissivity. As such, the algorithm theoretical basis seems to be badly flawed, which is a primary reason I cannot recommend this paper for publication at this time. The authors need to provide a much more convincing argument for the theoretical basis. The generation of the various empirical relationships is also poorly explained. The term "reemitted" is used. I recommend not using this term as matter emits radiation because it has a temperature. Once a photon is absorbed it should be considered dead and gone. Even though the algorithm is restricted to pixels that meet certain BTD requirements thought to be related to optical depth the background atmosphere and surface and viewing angle will still influence the retrieval to varying degrees. The authors should include a sensitivity analysis that justifies their assumptions, as most modern retrieval methods do not make such assumptions.*

There is a typing mistake in the referenced section. Instead of $Q_{ext} \; \alpha \; \frac{1}{I_0}$ , it should be $Q_{ext} \; \alpha \; (1 - \frac{I_r}{I_0}) \; (from \; Q_{ext} = \frac{P_{removed}}{I_0})$ where $(I_r)$ is the radiance received by satellite radiometer. The pretext in the introduction and the context that follows the relationships fits this intention. However, the Authors agree that there is over simplification in the analogy of the radiative transfer in *page 4* because of using Rayleigh-Jeans law which is not appropriate in thermal infrared part of the spectrum.

The authors acknowledge the complexity of the retrieving effective dust particle size using an analytical approach. Apart from the recent improvement, many studies tackled this problem through theoretical analysis but had limited success in filling the gap

between the observed and the modeled particle size has been achieved to date. The reason, as the reviewer points out, is the high number of dependent variables that link the remotely sensed radiance and particle size in the radiative transfer theory. The high uncertainty in the approximation of variables such as the chemical composition and particle shape, might be one reason to limit the advance in improving the accuracy of retrievals. The results of previous particle size models seem to inherit the noise introduced by the vague estimation of the many dependent variables. In a single thermal SEVIRI band, the effect of dust diameter is potentially "diluted" and difficult to see while the case turn out to be different in Brightness Temperature Difference of 8.7 and 12.0$\mu$m ($\Delta T_{8-12}$). This study approach aim to avoid the inherited noise of many dependent variables by exploiting the strong and dominate exponential effect of the particle size on the value of $\Delta T_{8-12}$. Here we try to present the empirical evidence of this relation and then use it to build a formula based mainly on empirical data and a simplified conceptual model.

**Action:** To make the thesis of the paper clearer, Section 3 has been rewritten to include more details explaining the empirical evidence of the dominant relation between $\Delta T_{8-12}$ and effective diameter. More detailed description of the effect of surface emissivity, water vapour, and dust has been also included in section 3. There are also more details for the theoretical basis in section 2. In section 4.3(Discussion of Results) more details discussion on the limitation of the model has been presented.
The word reemitted has been changed.